# Heart rate variability in hypothyroid patients: A systematic review and meta-analysis

**Valentin Brusseau**[1]*, **Igor Tauveron**[2], **Reza Bagheri**[3], **Ukadike Chris Ugbolue**[4], **Valentin Magnon**[5], **Valentin Navel**[6], **Jean-Baptiste Bouillon-Minois**[7], **Frederic Dutheil**[8]

**1** CHU Clermont–Ferrand, Endocrinology Diabetology and Metabolic Diseases, University Hospital of Clermont–Ferrand, Clermont-Ferrand, France, **2** GReD, CNRS, INSERM, University Hospital of Clermont–Ferrand, CHU Clermont–Ferrand, Endocrinology Diabetology and Metabolic Diseases, University of Clermont Auvergne, Clermont–Ferrand, France, **3** Exercise Physiology, University of Isfahan, Isfahan, Iran, **4** University of the West of Scotland, Health and Life Sciences, Institute for Clinical Exercise & Health Science, University of Strathclyde, Glasgow, Scotland, United Kingdom, **5** CNRS, LaPSCo, Physiological and Psychosocial Stress, University of Clermont Auvergne, Clermont–Ferrand, France, **6** CNRS, INSERM, GReD, CHU Clermont-Ferrand, University Hospital of Clermont-Ferrand, Ophthalmology, University of Clermont Auvergne, Clermont-Ferrand, France, **7** CNRS, LaPSCo, Physiological and Psychosocial Stress, University Hospital of Clermont–Ferrand, CHU Clermont–Ferrand, Emergency Medicine, University of Clermont Auvergne, Clermont–Ferrand, France, **8** CNRS, LaPSCo, Physiological and Psychosocial Stress, University Hospital of Clermont–Ferrand, CHU Clermont–Ferrand, Occupational and Environmental Medicine, WittyFit, University of Clermont Auvergne, Clermont–Ferrand, France

* vbrusseau@chu-clermontferrand.fr

## Abstract

### Introduction

Hypothyroidism may be associated with changes in the autonomic regulation of the cardio-vascular system, which may have clinical implications.

### Objective

To conduct a systematic review and meta-analysis on the impact of hypothyroidism on HRV.

### Materials and methods

PubMed, Cochrane, Embase and Google Scholar were searched until 20 August 2021 for articles reporting HRV parameters in untreated hypothyroidism and healthy controls. Random-effects meta-analysis were stratified by degree of hypothyroidism for each HRV parameters: RR intervals (or normal to normal-NN intervals), SDNN (standard deviation of RR intervals), RMSSD (square root of the mean difference of successive RR intervals), pNN50 (percentage of RR intervals with >50ms variation), total power (TP), LFnu (low-frequency normalized unit), HFnu (high-frequency), VLF (very low frequency), and LF/HF ratio.

### Results

We included 17 studies with 11438 patients: 1163 hypothyroid patients and 10275 healthy controls. There was a decrease in SDNN (effect size = -1.27, 95% CI -1.72 to -0.83),

**Data Availability Statement:** All relevant data are within the paper and its Supporting Information files.

**Funding:** The authors received no specific funding for this work.

**Competing interests:** The authors have declared that no competing interests exist.

RMSSD (-1.66, -2.32 to -1.00), pNN50 (-1.41, -1.98 to -0.84), TP (-1.55, -2.1 to -1.00), HFnu (-1.21, -1.78 to -0.63) with an increase in LFnu (1.14, 0.63 to 1.66) and LF/HF ratio (1.26, 0.71 to 1.81) (p <0.001). HRV alteration increased with severity of hypothyroidism.

## Conclusions

Hypothyroidism is associated with a decreased HRV, that may be explained by molecular mechanisms involving catecholamines and by the effect of TSH on HRV. The increased sympathetic and decreased parasympathetic activity may have clinical implications.

## Introduction

The heart is richly innervated by vagal and sympathetic fibers and is sensitive to autonomic influences [1]. The autonomic nervous system, by its sympathetic and parasympathetic divisions, regulates and modulates involuntary body functions. Dysautonomia refers to a change in the function of the autonomic nervous system that negatively affects a person's health [2], including increased cardiovascular morbidity [3]. Thyroid insufficiency or hypothyroidism is the inability of the thyroid gland to produce enough thyroid hormone. It is the most common hormonal disorder with a prevalence of 4–9% in women and 1–3% in men [4, 5]. Clinical signs of hypothyroidism include cardiovascular signs (bradycardia, decreased cardiac output and cardiac contractility] and suggest hypoactivity of the sympathetic nervous system [6]. If undiagnosed or insufficiently supplemented, hypothyroidism may be associated with changes in the autonomic regulation of the cardiovascular system. Heart rate variability (HRV) consists of the measurement of the physiological variation of RR intervals, a simple and convincing diagnostic tool used to assess the cardiac component of the autonomic nervous system [7–10]. Low HRV is an independent predictor of cardiac morbidity [11], while high HRV suggests good ability to adapt and respond to internal and external stimuli [3, 12]. Many studies have evaluated HRV parameters in hypothyroidism, but the results remain contradictory [13–18], although all tend to express the existence of alterations in parasympathetic and sympathetic activities in hypothyroidism compared with healthy controls. Few studies have comprehensively evaluated the role of the most common variables, such as age, sex, body mass index (BMI), blood pressure or biochemical thyroid function on HRV parameters in hypothyroidism [19, 20]. Therefore, we aimed to conduct a systematic review and meta-analysis on the impact of untreated hypothyroidism on HRV parameters. A secondary objective was to identify the most frequently reported predictors.

## Methods

### Literature search

All studies measuring HRV in patients with untreated hypothyroidism and healthy controls were reviewed until August 20, 2021, on the major article databases (PubMed, Cochrane Library, Embase, and Google Scholar) with the following keywords: ("hypothyroidism" OR "hypothyroid") AND ("heart rate variability" OR "HRV"). We included all articles that met our inclusion criteria of measuring HRV parameters in hypothyroid patients and healthy controls, regardless of article language and year of publication. There were no restrictions on the regional origin or nature of the control group. We excluded studies evaluating the effect of treated hypothyroidism on HRV parameters, without HRV parameters in the time or

frequency domain, without a control group, on animals, on children, conferences, congresses, or seminars. Studies had to be primary research. We manually searched the reference lists of all publications with our inclusion criteria to identify studies that would not have been found in the electronic search. We also performed searches within references of included articles or review found using our search strategy, to identify other potentially eligible primary studies. Our search strategy is shown in Fig 1 and S1 Fig. Two authors (VB and RB) conducted the literature searches, reviewed the abstracts and articles independently, checked suitability for inclusion, and extracted the data. When necessary, disagreements were solved with a third author (FD).

## Data extraction

The primary endpoint was the analysis of HRV parameters in untreated hypothyroid patients and in healthy controls. Linear methods are the most traditional measurement of HRV, including time and frequency domains [3]. In the time domain, the RR intervals (or normal-to-normal intervals-NN), the standard deviation of RR intervals (SDNN), the root mean square of successive RR-intervals differences (RMSSD) and the percentage of adjacent NN intervals varying by more than 50 milliseconds (pNN50) were analysed. The frequency domain can be separated in three components according to its frequency ranges [3]: low frequency (LF, 0.04 to 0.15 Hz), high frequency (HF, 0.15 to 0.4 Hz), and very low frequency (VLF, 0.003 to 0.04 Hz). Power is the energy found in a frequency band [21]. LF, HF, and VLF bands are obtained either with the fast Fourier transform algorithm or with autoregressive modelling [3]. LF and HF powers are absolute powers, reported in units of $ms^2$ (square milliseconds). LFnu and HFnu are relative power, called normalized power, in the LF and HF bands, a derived index that is calculated by dividing LF or HF by an appropriate denominator representing the relevant total power: LFnu = LF / (LF + HF) and HFnu = HF / (LF + HF). Due to high inter-individual variability in total and specific band power, LFnu and HFnu allow comparison of frequency domain HRV parameters between two patients [22]. RMSSD and pNN50 are associated with HF and HFnu power, which represents parasympathetic activity, whereas SDNN is associated with LF power, which represents both sympathetic and parasympathetic activity [23]. LFnu emphasizes the control and balance of cardiac sympathetic behaviour [24]. VLF power is also correlated with SDNN measurement due to still uncertain physiological mechanisms [25], thus both sympathetic and parasympathetic activity contribute to VLF power [26, 27]. Total power (TP) and LF/HF ratio, which represented sympathovagal balance, were calculated and reported in this meta-analysis. Secondary outcomes included clinical (BMI, blood pressure, treatments, other diseases), electrical (heart rate), hypothyroidism (duration, etiology, thyroid-stimulating hormone–TSH, free thyroxine–fT4, free triiodothyronine–fT3) and sociodemographic (age, sex, smoking) characteristics (Table 1).

## Quality of assessment

We used the Scottish Intercollegiate Guidelines Network (SIGN) score, based on different evaluation grids depending on the type of study. For cohort and cross-sectional studies, the evaluation grids are composed of two sections with 4 possible answers (yes, no, can't say or not applicable): one on the design of the study (14 items) and the other on the overall evaluation of the article (3 items) (S2 Fig) [28]. The "STrengthening the Reporting of OBservational studies in Epidemiology" (STROBE) score is used to check the quality of reports from cohort and cross-sectional studies [29]. By assigning one point per item and subitem, we were able to calculate a percentage of a maximum score of 32 points.

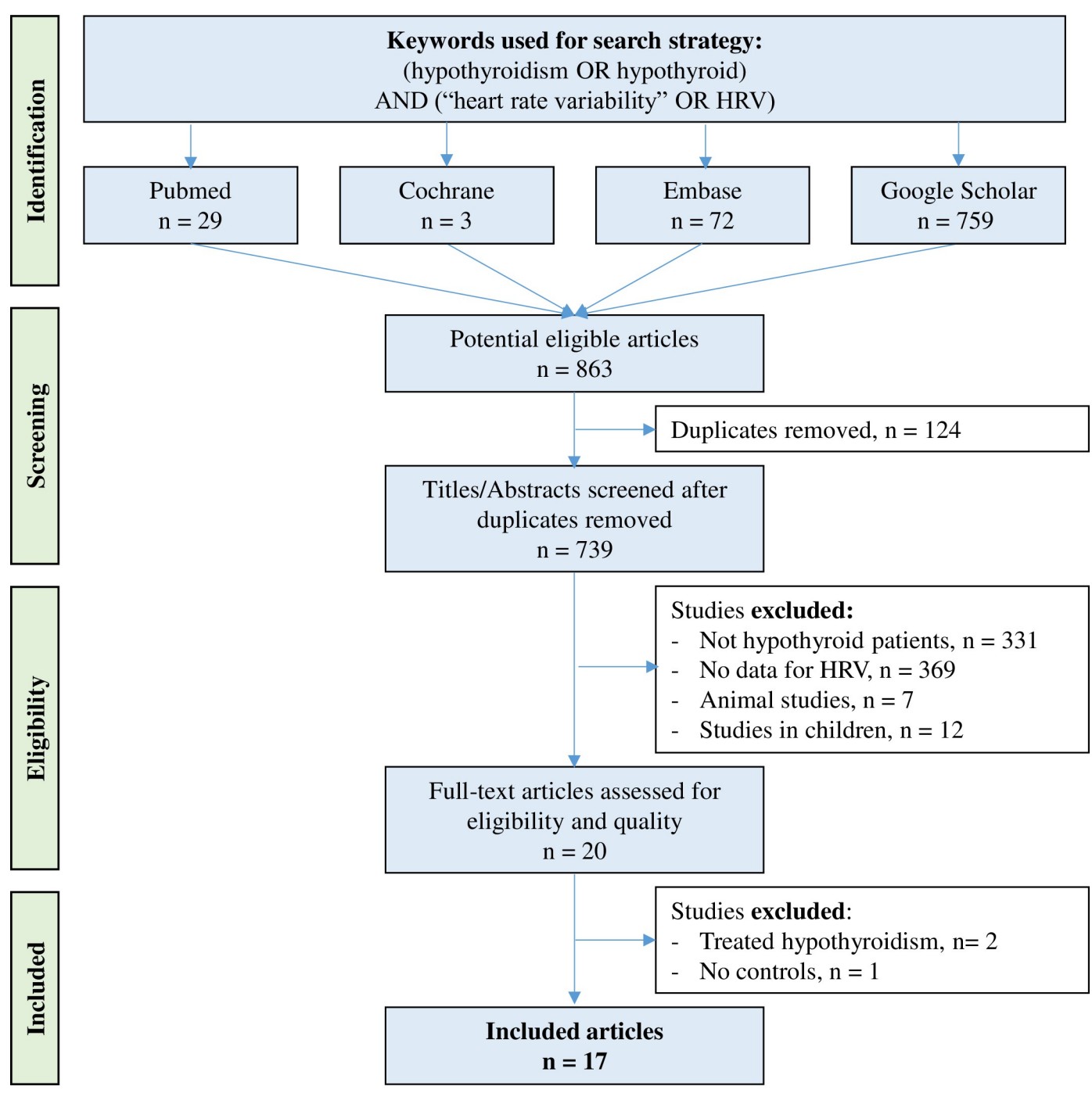

HRV: Heart rate variability

**Fig 1. Flow chart.**

## Statistical considerations

We used Stata software (v16, StataCorp, College Station, US) for the statistical analysis [30–34]. Main characteristics were synthetized for each study population and reported as

**Table 1. Descriptive characteristics of HRV parameters.**

| HRV parameters | | |
|---|---|---|
| Acronym (unit) | Full name | Signification |
| **Time-domain** | | |
| RR (ms) | RR–intervals (or Normal to Normal intervals–NN) i.e. beat-by-beat variations of heart rate | Overall autonomic activity |
| SDNN (ms) | Standard deviation of RR intervals | Correlated with LF power |
| RMSSD (ms) | Root mean square of successive RR-intervals differences | Associated with HF power and hence parasympathetic activity |
| pNN50 (%) | Percentage of adjacent NN intervals varying by more than 50 milliseconds | Associated with HF power and hence parasympathetic activity |
| **Frequency-domain** | | |
| TP ($ms^2$) | Total power i.e. power of all spectral bands | Overall autonomic activity |
| VLF ($ms^2$) | Very Low Frequency (0.003 to 0.04 Hz) | Thermoregulation, renin-angiotensin system |
| LF ($ms^2$) | Power of the high-frequency band (0.04–0.15 Hz) | Index of both sympathetic and parasympathetic activity, with a predominance of sympathetic |
| HF ($ms^2$) | Power of the high-frequency band (0.15–0.4 Hz) | Represents the most efferent vagal (parasympathetic) activity to the sinus node |
| LF/HF | LF/HF ratio | Sympathovagal balance |

mean ± standard deviation (SD) for continuous variables and number (%) for categorical variables. When data could be pooled, we conducted random effects meta-analyses (DerSimonian and Laird approach) for each HRV parameter comparing patients with untreated hypothyroidism with healthy controls [35]. A negative effect size (ES, standardised mean differences—SMD) [36] denoted lower HRV in patients than in controls. An ES is a unitless measure, centred at zero if the HRV parameter did not differ between hypothyroidism patients and controls. An ES of -0.8 reflects a large effect i.e. a large HRV decrease in patients compared to controls, -0.5 a moderate effect, and -0.2 a small effect. Then, meta-analyses stratified on TSH levels (above and below 10mIU/L or undefined if the TSH level was missing) were performed. We evaluated heterogeneity in the study results by examining forest plots, confidence intervals (CI) and I-squared ($I^2$). $I^2$ is the most common metric to measure heterogeneity between studies, ranging from 0 to 100%. Heterogeneity is considered low for $I^2<25\%$, modest for $25<I^2<50\%$, and high for $I^2>50\%$. We also searched for potential publication bias by examining funnel plots of these meta-analyses. We verified the strength of our results by conducting further meta-analyses after exclusion of studies that were not evenly distributed around the base of the funnel. If the sample size was sufficient, meta-regressions were performed to investigate the relationship between each HRV parameter and relevant clinicobiological parameters (age, sex, blood pressure, BMI, TSH, fT4 levels, fT3 levels). Results were expressed as regression coefficients and 95% confidence intervals (95%CI). P-values less than 0.05 were considered statistically significant.

## Results

An initial search produced a possible 863 articles (Fig 1). The number of articles reporting the evaluation of HRV in untreated hypothyroidism was reduced to 17 after elimination of duplicates and use of the selection criteria [15–17, 37–48]. All included articles were written in English.

Among the 17 studies included, six studies were prospective [16, 17, 40–42, 44], nine were cross-sectional [15, 37, 38, 43, 45–50] and one was retrospective [39]. Included studies were

published from 2000 to 2018 and conducted across 3 continents (Asia– 8 studies, Europe– 7 studies, America– 2 studies). All included articles compared HRV parameters of patients with untreated hypothyroidism and healthy controls [15–17, 37–48].

Sample size ranged from 14 [16] to 9134 [39], for a total of 11438 patients: 1163 with untreated hypothyroidism and 10275 healthy controls.

Thyroid function was described clinically and biologically in all studies. TSH levels was reported in all studies except two [43, 50]. Nine articles included hypothyroid patients with TSH >10mIU/L [15–17, 37, 38, 42, 44, 47, 48], five with TSH <10mIU/L [39–41, 45, 49], and one with both [46]. Most studies included newly diagnosed and untreated hypothyroid patients before initiation of therapy [16, 17, 37, 38, 43, 47, 48].

HRV recording was ambulatory, spontaneous breathing with normal daily activity in all studies. Most studies used ECG in the supine position at rest to determine HRV [15, 16, 37–39, 43–45, 47, 48, 50], ranging from 4 [37] to 15 minutes [44], except six studies that used a 24-hour holter-ECG [17, 40–42, 46, 49]. Parameters reported were both time and frequency domains in most studies, except two studies that reported only time domain [40, 49] and one only frequency domain [15].

More details on study characteristics (Table 2), aims and quality of articles, inclusion and exclusion criteria, characteristics of population, characteristics of hypothyroidism, and HRV measurements and analysis are described in S3 Fig.

## Meta–analyses of HRV values in untreated hypothyroidism

The main results of the meta-analysis are shown in Fig 2. In comparison to healthy controls, we noted strong evidence (p <0.001) that hypothyroid patient had significantly lower SDNN (ES = -1.27, 95% CI -1.72 to -0.83), RMSSD (-1.66, -2.32 to -1.00), pNN50 (-1.41, -1.98 to -0.84), TP (-1.55, -2.1 to -1.00), LF power (-0.58, -0.89 to -0.28), HF power (-0.98, -1.44 to -0.51), HFnu (-1.21, -1.78 to -0.63) and higher LFnu (1.14, 0.63 to 1.66) and LF/HF ratio (1.26, 0.71 to 1.81). There was no significant difference in RR intervals between hypothyroid patients and healthy controls (p = 0.174) (S4 Fig).

## Meta-analysis stratified by TSH levels

RR intervals and LF/HF were only altered in the most severe patients (TSH >10mIU/L) (ES = 0.53, 95% CI 0.09 to 0.96 and 1.34, 0.69 to 2.00, respectively), and not when TSH levels were <10mIU/L (-0.72, -1.52 to 0.07 and 0.56, -0.29 to 1.41, respectively). Despite non-significant comparisons between subgroups, we noted a global higher decrease in HRV when TSH was >10mIU/L: SDNN (-1.17, -1.63 to -0.70 for TSH>10mIU/L and -0.77, -1.23 to -0.31 for TSH<10mIU/L subgroup), RMSSD (-1.13, -1.84 to -0.43 and -1.49, -2.49 to -0.48), pNN50 (-1.19, -1.75 to -0.64 and -0.73, -1.43 to -0.03), LF power (-0.97, -1.68 to -0.25 and -0.35, -0.66 to -0.05) and HF power (-1.02, -1.8 to -0.26 and -0.96, -1.68 to -0.25) (p <0.05). Other parameters were only measured in the most severe patients (TSH >10mIU/L), precluding comparisons between the two subgroups based on TSH levels. However, they were strongly altered (ES greater than 0.80 or -0.80) in those severe patients: TP (-1.70, -2.32 to -1.07), and HFnu (-1.37, -2.01 to -0.73) and higher LFnu (1.28, 0.73 to 1.83) (S4 Fig). All meta-analyses had a high degree of heterogeneity ($I^2$>50%).

## Meta–regressions and sensitivity analyses

An increase in fT3 was associated with lower RR intervals (coefficient = -0.75, 95%CI -1.44 to -0.07) (p <0.05). Age was associated with lower RMSSD (-0.09, -0.17 to -0.004) (p = 0.041). Men had lower LFnu (-4.36, -8.53 to -0.19, per % men) and LF/HF (-6.08, -9.52 to -2.64) (p

**Table 2. Characteristics of included studies.**

| Study | Country | Design | Subgroup | Untreated hypothyroidism | | | | | | Healthy controls | | | ECG, min | HRV parameters |
|---|---|---|---|---|---|---|---|---|---|---|---|---|---|---|
| | | | | n | Age, years | Sex, % men | FT4, pmol/L | FT3, pmol/L | TSH, mIU/L | n | Age, years | Sex, % men | | |
| **Ahmed 2010** | Bangladesh | Cross-sectional | Overt | 30 | 38.0 ± 1.2 | 0.0% | 5.1 ± 1.9 | - | 38.2 ± 30.5 | 30 | 36.0 ± 2.6 | 0.0% | 5 | TP, LF, HF, LF/HF |
| **Cacciatori 2000** | Italy | Prospective | Lying–overt / Standing–overt | 7 | 52.1 ± 5.3 | 0.0% | 3.1 ± 0.4 | - | 55.5 ± 3.5 | 7 | 52.0 ± 5.2 | 0.0% | 10 | RR, TP, LF, HF, LF/HF |
| **Celik 2011** | Turkey | Prospective | Subclinical | 40 | 48.0 ± 13.0 | 10.0% | 11.6 ± 3.9 | 4.0 ± 1.1 | 6.2 ± 1.2 | 31 | 51.0 ± 12.0 | 9.7% | 1440 | RR, SDNN, RMSSD |
| **Falcone 2014** | Italy | Cross-sectional | Subclinical | 55 | 71.0 ± 13.1 | 23.6% | 24.5 ± 9.0 | 4.0 ± 1.2 | 5.4 ± 1.4 | 170 | 71.0 ± 12.4 | 34.7% | 1440 | RR, SDNN, RMSSD, pNN50 |
| **Galetta 2006** | Italy | Prospective | Subclinical | 42 | 53.2 ± 14.2 | 0.0% | 9.3 ± 1.1 | 4.3 ± 0.2 | 9.8 ± 1.7 | 30 | 51.4 ± 16.2 | 30.0% | 1440 | RR, SDNN, RMSSD, pNN50, LF, HF, LF/HF |
| **Galetta 2008** | Italy | Prospective | Overt | 31 | 53.6 ± 11.8 | 29.0% | 0.7 ± 0.1 | 1.8 ± 0.3 | 56.2 ± 14.7 | 31 | 50.4 ± 15.3 | 29.0% | 1440 | RR, SDNN, RMSSD, pNN50, LF, HF, LF/HF |
| **Gupta 2017** | Nepal | Cross-sectional | Subclinical | 30 | 32.0 ± 9.1 | 33.3% | - | - | 22.8 ± 3.5 | 30 | 29.3 ± 6.2 | 33.3% | 5 | SDNN, RMSSD, pNN50, TP, LF, HF |
| **Heemstra 2010** | The Netherlands | Prospective | Overt | 11 | 45.5 ± 10.0 | 36.4% | 1.4 ± 0.7 | 0.1 ± 0.2 | 142.4 ± 34.4 | 21 | 45.5 ± 8.7 | 38.1% | 15 | RR, LF, HF, VLF, LF/HF |
| **Hoshi 2018** | Brazil | Cross-sectional | Subclinical / Overt | 44 / 59 | 55.0 ± 4.0 / - | 40.9% / - | 14.2 ± 1.3 / - | 4.9 ± 0.4 / - | 4.8 ± 1.0 / 8.7 ± 3.2 | 509 | 52.0 ± 6.5 | 56.6% | 10 | SDNN, RMSSD, pNN50, LF, HF, LF/HF |
| **Karthik 2009** | India | Cross-sectional | Overt | 15 | 29.2 ± 5.7 | 0.0% | 4.0 ± 1.7 | 2.2 ± 0.8 | 88.5 ± 20.3 | 15 | 27.8 ± 6.6 | 0.0% | 4 | RR, SDNN, RMSSD, TP, LF, HF, LF/HF |
| **Mavai 2018** | India | Cross-sectional | Overt | 35 | 37.3 ± 9.3 | - | 9.0 ± 3.7 | 2.6 ± 1.0 | 16.9 ± 7.4 | 25 | 34.5 ± 10.1 | - | 5 | SDNN, RMSSD, pNN50, TP, LF, HF |
| **Moldabek 2011** | Kazakhstan | Cross-sectional | Overt | 42 | - | - | - | - | 32.0 ± 10.2 | 30 | - | - | 5 | RR, SDNN, RMSSD, pNN50, LF/HF |
| **Peixoto de Miranda 2018** | Brazil | Retrospective | Subclinical | 511 | 52.0 ± 6.5 | 47.2% | - | - | 5.1 ± 1.0 | 8623 | 50.0 ± 6.0 | 48.4% | 10 | RR, SDNN, RMSSD, pNN50, LF, HF |
| **Sahin 2005** | Turkey | Cross-sectional | Subclinical (TSH 4.4–9.9mIU/L) / Subclinical (TSH>10mIU/L) | 18 / 13 | 41.1 ± 12.6 / 41.1 ± 12.6 | 11.1% / 7.7% | - / - | - / - | 7.2 ± 3.9 / 20.6 ± 9.1 | 28 | 41.1 ± 15.2 | 7.1% | 1440 | SDNN, RMSSD, pNN50, LF, HF, LF/HF |
| **Syamsunder 2013** | India | Cross-sectional | Overt | 54 | 27.2 ± 4.7 | 0.0% | 8.0 ± 3.6 | 2.3 ± 0.8 | 97.6 ± 55.8 | 50 | 25.5 ± 5.6 | 0.0% | 10 | RR, SDNN, RMSSD, pNN50, TP, LF, HF, LF/HF |
| **Syamsunder 2016** | India | Cross-sectional | Subclinical | 81 | 27.3 ± 3.2 | 0.0% | 15.4 ± 6.6 | 4.1 ± 1.3 | 12.7 ± 2.3 | 80 | 36.6 ± 4.8 | 0.0% | 10 | RR, SDNN, RMSSD, pNN50, TP, LF, HF, LF/HF |

*(Continued)*

**Table 2.** (Continued)

| Study | Country | Design | Subgroup | Untreated hypothyroidism | | | | | | Healthy controls | | | ECG, min | HRV parameters |
|---|---|---|---|---|---|---|---|---|---|---|---|---|---|---|
| | | | | n | Age, years | Sex, % men | FT4, pmol/L | FT3, pmol/L | TSH, mIU/L | n | Age, years | Sex, % men | | |
| **Xing 2001** | China | Prospective | Overt | 38 | 51.0 ± 13.0 | 23.7% | 0.2 ± 0.1 | 0.9 ± 0.1 | 65.0 ± 25.6 | 21 | 52.0 ± 11.0 | 23.8% | 1440 | SDNN, RMSSD, pNN50, LF, HF, LF/HF |

FT4: free thyroxine, FT3: free triiodothyronine, TSH: thyroid-stimulating hormone, RR: RR intervals (or normal-to-normal intervals-NNs), SDNN: standard deviation of RR intervals, pNN50: percentage of adjacent NN intervals differing by more than 50 milliseconds, RMSSD: the square root of the mean squared difference of successive RR-intervals, TP: total power, LF: low frequency, HF: high frequency, VLF: very low frequency, LF/HF ratio: low frequency / high frequency ratio.

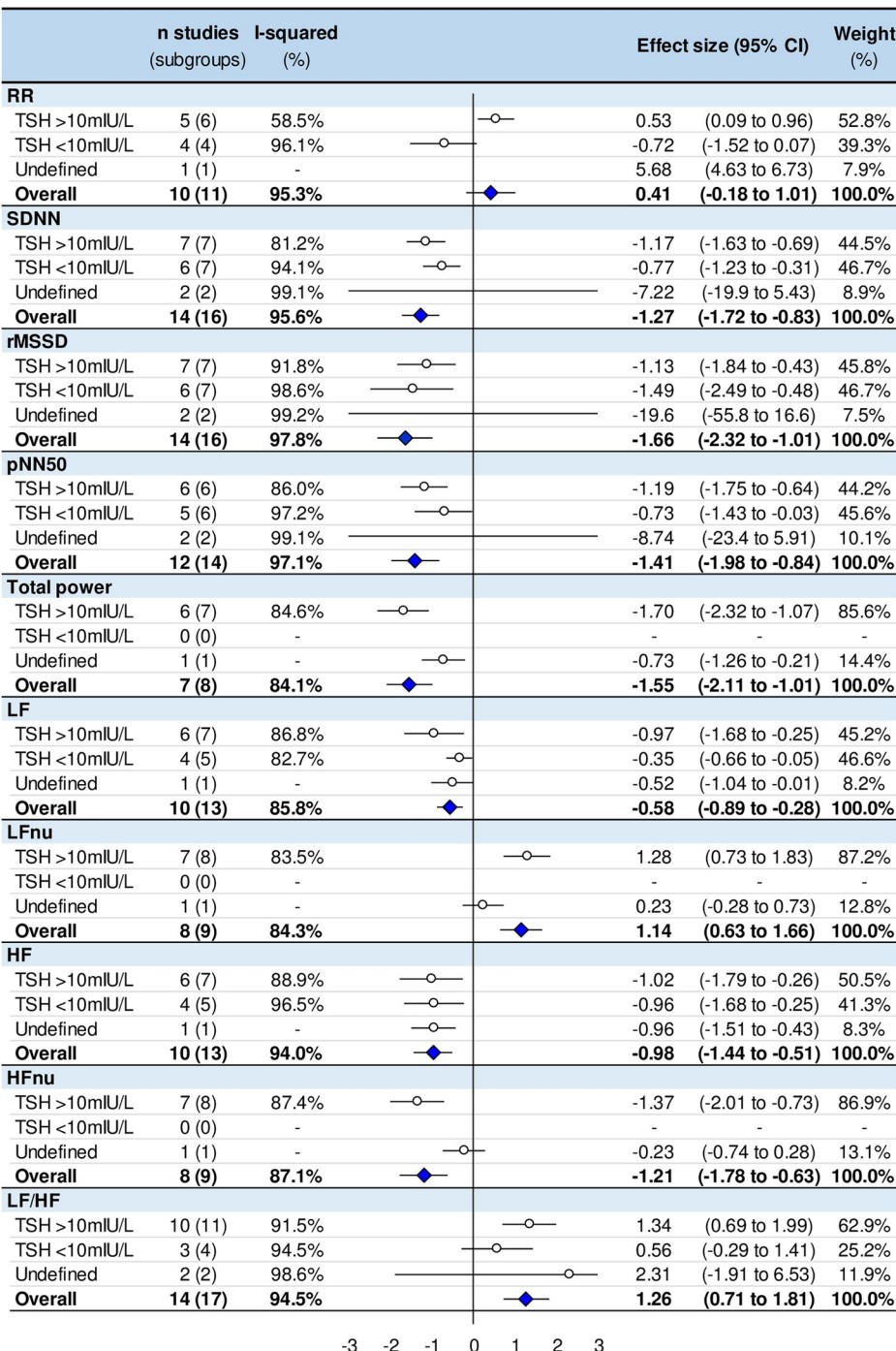

| | n studies (subgroups) | I-squared (%) | | Effect size (95% CI) | | Weight (%) |
|---|---|---|---|---|---|---|
| **RR** | | | | | | |
| TSH >10mIU/L | 5 (6) | 58.5% | | 0.53 | (0.09 to 0.96) | 52.8% |
| TSH <10mIU/L | 4 (4) | 96.1% | | -0.72 | (-1.52 to 0.07) | 39.3% |
| Undefined | 1 (1) | - | | 5.68 | (4.63 to 6.73) | 7.9% |
| **Overall** | **10 (11)** | **95.3%** | | **0.41** | **(-0.18 to 1.01)** | **100.0%** |
| **SDNN** | | | | | | |
| TSH >10mIU/L | 7 (7) | 81.2% | | -1.17 | (-1.63 to -0.69) | 44.5% |
| TSH <10mIU/L | 6 (7) | 94.1% | | -0.77 | (-1.23 to -0.31) | 46.7% |
| Undefined | 2 (2) | 99.1% | | -7.22 | (-19.9 to 5.43) | 8.9% |
| **Overall** | **14 (16)** | **95.6%** | | **-1.27** | **(-1.72 to -0.83)** | **100.0%** |
| **rMSSD** | | | | | | |
| TSH >10mIU/L | 7 (7) | 91.8% | | -1.13 | (-1.84 to -0.43) | 45.8% |
| TSH <10mIU/L | 6 (7) | 98.6% | | -1.49 | (-2.49 to -0.48) | 46.7% |
| Undefined | 2 (2) | 99.2% | | -19.6 | (-55.8 to 16.6) | 7.5% |
| **Overall** | **14 (16)** | **97.8%** | | **-1.66** | **(-2.32 to -1.01)** | **100.0%** |
| **pNN50** | | | | | | |
| TSH >10mIU/L | 6 (6) | 86.0% | | -1.19 | (-1.75 to -0.64) | 44.2% |
| TSH <10mIU/L | 5 (6) | 97.2% | | -0.73 | (-1.43 to -0.03) | 45.6% |
| Undefined | 2 (2) | 99.1% | | -8.74 | (-23.4 to 5.91) | 10.1% |
| **Overall** | **12 (14)** | **97.1%** | | **-1.41** | **(-1.98 to -0.84)** | **100.0%** |
| **Total power** | | | | | | |
| TSH >10mIU/L | 6 (7) | 84.6% | | -1.70 | (-2.32 to -1.07) | 85.6% |
| TSH <10mIU/L | 0 (0) | - | | - | - | - |
| Undefined | 1 (1) | - | | -0.73 | (-1.26 to -0.21) | 14.4% |
| **Overall** | **7 (8)** | **84.1%** | | **-1.55** | **(-2.11 to -1.01)** | **100.0%** |
| **LF** | | | | | | |
| TSH >10mIU/L | 6 (7) | 86.8% | | -0.97 | (-1.68 to -0.25) | 45.2% |
| TSH <10mIU/L | 4 (5) | 82.7% | | -0.35 | (-0.66 to -0.05) | 46.6% |
| Undefined | 1 (1) | - | | -0.52 | (-1.04 to -0.01) | 8.2% |
| **Overall** | **10 (13)** | **85.8%** | | **-0.58** | **(-0.89 to -0.28)** | **100.0%** |
| **LFnu** | | | | | | |
| TSH >10mIU/L | 7 (8) | 83.5% | | 1.28 | (0.73 to 1.83) | 87.2% |
| TSH <10mIU/L | 0 (0) | - | | - | - | - |
| Undefined | 1 (1) | - | | 0.23 | (-0.28 to 0.73) | 12.8% |
| **Overall** | **8 (9)** | **84.3%** | | **1.14** | **(0.63 to 1.66)** | **100.0%** |
| **HF** | | | | | | |
| TSH >10mIU/L | 6 (7) | 88.9% | | -1.02 | (-1.79 to -0.26) | 50.5% |
| TSH <10mIU/L | 4 (5) | 96.5% | | -0.96 | (-1.68 to -0.25) | 41.3% |
| Undefined | 1 (1) | - | | -0.96 | (-1.51 to -0.43) | 8.3% |
| **Overall** | **10 (13)** | **94.0%** | | **-0.98** | **(-1.44 to -0.51)** | **100.0%** |
| **HFnu** | | | | | | |
| TSH >10mIU/L | 7 (8) | 87.4% | | -1.37 | (-2.01 to -0.73) | 86.9% |
| TSH <10mIU/L | 0 (0) | - | | - | - | - |
| Undefined | 1 (1) | - | | -0.23 | (-0.74 to 0.28) | 13.1% |
| **Overall** | **8 (9)** | **87.1%** | | **-1.21** | **(-1.78 to -0.63)** | **100.0%** |
| **LF/HF** | | | | | | |
| TSH >10mIU/L | 10 (11) | 91.5% | | 1.34 | (0.69 to 1.99) | 62.9% |
| TSH <10mIU/L | 3 (4) | 94.5% | | 0.56 | (-0.29 to 1.41) | 25.2% |
| Undefined | 2 (2) | 98.6% | | 2.31 | (-1.91 to 6.53) | 11.9% |
| **Overall** | **14 (17)** | **94.5%** | | **1.26** | **(0.71 to 1.81)** | **100.0%** |

-3   -2   -1   0   1   2   3

*RR: RR intervals (or normal-to-normal intervals-NNs), SDNN: standard deviation of RR intervals, pNN50: percentage of adjacent NN intervals differing by more than 50 milliseconds, RMSSD: the square root of the mean squared difference of successive RR-intervals, LF: low frequency, LFnu: low frequency normalized – units, HF: high frequency, HFnu: high frequency – normalized units, LF/HF ratio: low frequency / high frequency ratio*

**Fig 2. Meta-analysis of heart rate variability parameters of untreated hypothyroid patients compared with controls.**

| | n subgroups | I-squared (%) | | Coefficient (95% CI) | p-value |
|---|---|---|---|---|---|
| **RR** | | | | | |
| FT3, pmol/L | 7 | 99.00% | | -0.75 (-1.44 to -0.07) | **0.037** |
| **RMSSD** | | | | | |
| Age, years | 12 | 98.20% | | -0.09 (-0.17 to -0.01) | **0.041** |
| **LF** | | | | | |
| DBP, mmHg | 6 | 26.50% | | -0.25 (-0.45 to -0.06) | **0.023** |
| **LFnu** | | | | | |
| Sex, %men | 8 | 78.60% | | -4.36 (-8.53 to -0.19) | **0.043** |
| **HFnu** | | | | | |
| SBP, mmHg | 7 | 80.80% | | -0.08 (-0.15 to -0.01) | **0.030** |
| **LF/HF** | | | | | |
| Sex, %men | 14 | 85.20% | | -6.08 (-9.52 to -2.64) | **0.002** |

```
        -9    -6    -3     0     3
```

*RR: RR intervals (or normal-to-normal intervals-NNs), BMI: body mass index, FT4: free thyroxine, FT3: free triiodothyronine, TSH: thyroid-stimulating hormone, RMSSD: the square root of the mean squared difference of successive RR-intervals, SBP: systolic blood pressure, VLF: very low frequency, LF: low frequency, LFnu: low frequency – normalized units, HF: high frequency, HFnu: high frequency – normalized units, LF/HF ratio: low frequency / high frequency ratio*

**Fig 3. Meta-regressions of significant factors influencing heart rate variability in untreated hypothyroid patients (exhaustive metaregressions are presented in S5 Fig).**

<0.05). An increase in systolic blood pressure was associated with lower HFnu (-0.08, -0.15 to -0.01) and an increase in diastolic blood pressure was associated with lower LF power (-0.25, -0.45 to -0.06) (p <0.05). No significant results were observed for BMI, fT4 and TSH levels (Fig 3 and S5 Fig).

The meta-analyses were rerun after excluding studies that were not evenly distributed around the base of the funnel (S6 Fig) and showed similar results.

## Discussion

The main results showed a decreased HRV in patients with hypothyroidism that may be explained by the deleterious effect of TSH. The increase in sympathetic and decrease in parasympathetic activity may have clinical implications. Some other factors, such as age or BMI, should also be considered from a clinical perspective.

### Deleterious effects of hypothyroidism on HRV

Hypothyroidism is often considered to influence the autonomic nervous system in the opposite direction to hyperthyroidism [51]. Based on clinical data, a decrease in sympathetic activity would be suggested [16]. However, production, release and plasma degradation of catecholamines is increased in hypothyroidism, explaining increased sympathetic activity [14, 52]. These data suggest desensitisation of catecholamine receptors or post-receptor sites in hypothyroidism [16, 53, 54], with reduced binding of β- and α2-adrenergic receptors in cardiac myocytes [53, 54]. These results are consistent with the increased muscle sympathetic activity

in hypothyroidism [55]. Similarly, the decreased parasympathetic activity in hypothyroidism may be explained by neuroterminal alteration of cardiac parasympathetic neurons and thus, a decrease in muscarinic effect [56, 57]. Vagal inhibition is more intense than increased sympathetic activity, with a greater decrease in HF power than LF power. Logically, TP decreases markedly (cardiac vagal control) as HF is its main contributor–two third, while LF and VLF contributes one third [3, 58]. HRV is decreased mainly because of a large decrease in vagal activity [3, 58]. No differences in RR intervals is common in hypothyroidism [59], this is in line with our results. The hypothalamus is involved in cardiac autonomic control and TSH release [60, 61], linking the thyroid to the autonomic nervous system [62, 63]. In hypothyroidism, the cardiac autonomic alteration may take place at an hypothalamic level [64]. Indeed, some studies suggested that TSH stimulates sympathetic output from the central nervous system and acts as a neurotransmitter, playing a critical role in determining sympathovagal imbalance [65]. It corroborates the greater HRV decrease in patients with higher TSH levels [45, 46].

## Clinical implications

Decreased vagal tone and increased sympathetic activity in hypothyroidism have important clinical implications. Catecholamine receptor desensitization results in a decrease cardiac output, leading to a compensatory increase in norepinephrine release [66]. Hypothyroidism is associated with an increased risk of cardiovascular mortality [67], coronary artery disease [49], and potentially fatal arrhythmias [68, 69]. These complications result from multiple mechanisms (reduced systolic function, diastolic hypertension, atherogenic profile), but also sympathovagal imbalance [41, 69]. Indeed, patients with low vagal tone are more susceptible to cardiovascular diseases such as myocardial infarction, rhythm disorders, and hypertension [70, 71]. It has also been shown that decreased TP predicts an increased risk of sudden cardiac death [72] and total cardiac mortality [73], and that decreased LF was a strong predictor of sudden death independently of other variables [74]. These data suggest that HRV parameters may be a marker of increased mortality in hypothyroid patients [40]. The cardiac effects of hypothyroidism depend on the severity of the disease [65], with higher TSH levels associated with a higher risk of sudden cardiac death [75]. Therefore, it may be worthwhile to consider treatment of hypothyroidism, even for TSH <10mIU/L. However, reversibility of HRV abnormalities in hypothyroidism is not yet demonstrated to prevent cardiac complications.

## Other variables related to HRV in hypothyroidism

An increase in fT3 was associated with lower RR, which seems logical as thyroid hormones increase intrinsic activity of the sinus node and thus heart rate [76]. Men were associated with lower LF/HF ratio. This may be explained by the fact that men have lower sympathetic activity and higher parasympathetic activity compared to women [77], hence a decrease in LF/HF ratio [78, 79]. The sympathovagal imbalance could be due to a change in lipid profile as dyslipidemia is common in hypothyroidism [6], and is associated with increased sympathetic activity [80, 81]. However, this variable could not be explored in our meta-analysis due to lack of data. Age was associated with a decreased RMSSD. Indeed, the levels of the HRV time domain parameters decrease with age, especially after 50 years [82, 83] and the prevalence of hypothyroidism increases with age up to 10–15% in elderly patients [4]. We demonstrated that increased diastolic and systolic blood pressure were associated with decreased LF and HFnu power, respectively. The disturbance in blood pressure balance in hypothyroidism with systolic hypotension and diastolic hypertension, possibly reflects an alteration of the autonomic nervous system [84].

## Limitations

All meta-analyses have limitations, including those of the individual studies that comprise them, and are theoretically subjected to publication bias [85]. Although the meta-analysis was based on a moderate number of studies [86], the use of broader keywords in the search strategy limits the number of missing studies. The included studies were of variable quality despite our inclusion criteria [39, 50]. Most studies were cross-sectional [15, 37, 38, 43, 45–50], precluding robust conclusions for our meta-analyses [86]. Data collection, inclusion criteria and exclusion criteria were not identical in each study, although similar, which may have affected our results [87]. We limited the influence of extreme results and heterogeneity by exclusion of outliers [88, 89]. In addition, all studies except one [39] were monocentric, limiting the generalizability of our results [87]. Moreover, declarative data from studies are a putative bias [85]. Studies also differed in measurement conditions, such as in duration of recording of HRV parameters [38, 46]. No included studies assessed pulse-based HRV that seems to be less accurate than ECG-based HRV [90]. The interpretation of the LF/HF ratio is controversial in the literature, and may not correspond exactly to the sympatho-vagal balance [91, 92]. Ideally, the sympatho-vagal system tends more towards a non-linear relationship [91, 93]. We did not compute meta-analysis on non-linear assessment of HRV as it has been poorly studied in hypothyroidism. Parasympathetic-sympathetic interactions are complex, non-linear and often non-reciprocal [21]. Thus, non-linear measurements of HRV allow the unpredictability of a time series to be quantified [92], which results from the complexity of the HRV regulatory mechanisms [94–96]. Similarly, VLF power has been investigated by only one study [44] and is recognized as an independent predictor of mortality in patients with heart failure or in chronic hemodialysis patients [97]. The potential importance of VLF in hypothyroidism should be further investigated. Despite most included articles did not show HRV alteration depending on levels of TSH, we showed significant dose response relationship. It may be explained by the fact that each included article only retrieved a small increase in TSH levels, which may explain the absence of significant relation, whereas the combination of all articles in our meta-analysis permitted to analyze a wide range of TSH levels and HRV values. Etiology, duration of hypothyroidism and lipid profile were poorly reported, precluding further analysis. Similarly, the lack of data on spectral analysis of hypothyroidism with TSH below 10mIU/L did not allow conclusion on the type and degree of sympathovagal imbalance.

## Conclusion

HRV is markedly decreased in hypothyroid patients. Increased sympathetic and decreased parasympathetic activity may be explained by molecular mechanisms involving catecholamines and by the effect of TSH on HRV parameters. The increased sympathetic and decreased parasympathetic activity may have clinical implications.

## Supporting information

**S1 Checklist.**
(DOC)

**S1 Fig. Details for the search strategy used within each database.**
(TIFF)

**S2 Fig. Quality of included studies.** Methodological quality of included studies using the SIGN checklist. Methodological quality of included studies using the SIGN checklist, by study. SIGN checklist for cohort studies. Methodological quality of included studies using STROBE

checklist, by study.
(TIFF)

**S3 Fig. Aims of included articles, quality of articles, inclusion and exclusion criteria of included studies, characteristics of population, characteristics of hypothyroidism, and HRV measurements and analysis.**
(TIFF)

**S4 Fig. Detailed meta–analyses in untreated hypothyroid patients compared with controls for each HRV parameters: RR intervals, SDNN, RMSSD, pNN50, TP, LF, HF, LF/HF.**
(TIFF)

**S5 Fig. Detailed meta-regressions of factors influencing HRV parameters.**
(TIFF)

**S6 Fig. Meta funnels.**
(TIFF)

## Author Contributions

**Conceptualization:** Valentin Brusseau, Frederic Dutheil.

**Data curation:** Valentin Brusseau, Reza Bagheri.

**Formal analysis:** Valentin Navel, Frederic Dutheil.

**Investigation:** Valentin Navel.

**Methodology:** Valentin Brusseau, Valentin Navel, Jean-Baptiste Bouillon-Minois, Frederic Dutheil.

**Project administration:** Frederic Dutheil.

**Resources:** Valentin Magnon, Jean-Baptiste Bouillon-Minois.

**Software:** Valentin Magnon, Jean-Baptiste Bouillon-Minois, Frederic Dutheil.

**Supervision:** Igor Tauveron, Frederic Dutheil.

**Validation:** Igor Tauveron, Ukadike Chris Ugbolue, Frederic Dutheil.

**Visualization:** Igor Tauveron, Ukadike Chris Ugbolue, Frederic Dutheil.

**Writing – original draft:** Valentin Brusseau.

**Writing – review & editing:** Valentin Brusseau.

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
