## [Decision Letter · Decision Letter 0]

1 Apr 2022

PONE-D-22-02213Heart rate variability in hypothyroid patients: A systematic review and meta-analysisPLOS ONE

Dear Dr. Brusseau,

Thank you for submitting your manuscript to PLOS ONE. After careful consideration, we feel that it has merit but does not fully meet PLOS ONE’s publication criteria as it currently stands. Therefore, we invite you to submit a revised version of the manuscript that addresses the points raised during the review process. Please pay attention to all the reviewer's comments, especially those regarding potential bias and references and the use of digitized ECGs and HRVs from devices measuring it from a pulse wave based device. Use of the LF/HF ratio and current discussions regarding the utility of this should also be addressed.

We look forward to receiving your revised manuscript.

Kind regards,

Daniel M. Johnson, PhD

Academic Editor

PLOS ONE

Journal Requirements:

(This work was supported by the National Natural Science Foundation of China (Grant No. 51475317) and the Shanxi Provincial Natural Science Foundation of China (Grant No.201901D111237).)

(This work was supported by the National Natural Science Foundation of China (Grant No. 51475317) and the Shanxi Provincial Natural Science Foundation of China (Grant No.201901D111237).)

(This work was supported by the National Natural Science Foundation of China (Grant No. 51475317) and the Shanxi Provincial Natural Science Foundation of China (Grant No.201901D111237).)

5. Please amend the manuscript submission data (via Edit Submission) to author Bingbing Peng, and Pengcheng Liu.

6. Please amend your authorship list in your manuscript file to author Bingbing Peng, and Pengcheng Liu.

Reviewers' comments:

Reviewer's Responses to Questions

**Comments to the Author**

1. Is the manuscript technically sound, and do the data support the conclusions?

Reviewer #1: Partly

Reviewer #2: Partly

2. Has the statistical analysis been performed appropriately and rigorously? 

Reviewer #1: Yes

Reviewer #2: No

3. Have the authors made all data underlying the findings in their manuscript fully available?

Reviewer #1: Yes

Reviewer #2: Yes

4. Is the manuscript presented in an intelligible fashion and written in standard English?

Reviewer #1: Yes

Reviewer #2: Yes

5. Review Comments to the Author

Reviewer #1: This is an interesting systematic review and meta-analysis heart rate variability (HRV) in hypothyroid patients.

The conclusions from your review that HRV increases with severity of hypothyroidism are not particularly surprising. Much of the data from patients in the studies to which you refer were identified by routine blood test and were subclinical, their ‘hypothyroid status’ the result of deviation from the ‘normal values’.

Peixoto de Miranda et al., (2018) showed that subclinical hyperthyroidism leads to lower heart rate variability. These authors also found that no significant difference subclinical hypothyroidism group when compared to the euthyroid group. Peixoto de Miranda et al., (2018) therefore concluded that the subclinical thyroid dysfunctions presented no relationship with HRV variables. The conclusions from your paper seem to contradict this. Peixoto de Miranda et al., also adjusted for sociodemographic and clinical characteristics in their patients, and this is not apparent from the other references you have used in your review. Perhaps the negative findings from the Peixoto de Miranda et al. study are because the authors had adjusted for sociodemographic and clinical characteristics in their patients. It is well known that thyroid hormones and stress are linked in acute conditions.

In [line 384] you say that "Men were associated with lower LF/HF ratio. This may be explained by the fact that men have lower sympathetic activity and higher parasympathetic activity compared to women". My first question when reading this was the expectation that there would therefore be a sex difference in HRV and hypothyroidism. It is disappointing to see that there is insufficient data in your review to investigate this further.

You don't seem to query the standard interpretation of the LF/HF ratio as sympathetic/ parasympathetic. It is not unequivocally accepted by everyone. (Billman, GE., (2013) The LF/HF ratio does not accurately measure cardiac sympatho-vagal balance. Front. Physiol., 20 February 2013 | https://doi.org/10.3389/fphys.2013.00026).

[Line 425] Your conclusion is seriously deficient. Readers of this journal will be none the wiser by your statement about ‘deleterious effect of TSH on HRV parameters’. This is meaningless. There are a huge number of conditions that change HRV parameters that are not deleterious. Deleterious implies harm.

[Line 426] The statement ‘benefits of HRV assessment in the evaluation and monitoring of the severity of hypothyroidism should be further investigated’ is questionable – to what purpose? HRV can be affected by age, sex, stress, sociodemographic and clinical characteristics, and you have not given a breakdown due to lack of data.

I am happy with most of your interpretation of data. It is a shame that the lack of data precludes an adequate conclusion. It would help the reader if you were to include a list of abbreviations used in the paper. The world of HRV is difficult to understand by the novice.

Reviewer #2: The idea of doing a meta-analysis of HRV and hypothyroidism that includes both short term resting HRV and 24-hour recordings cannot be justified. SDNN from a 24-hour recording and SDNN from a 5-min recording cannot be compared and some of the short recordings even included paced breathing. SDNN is a powerful snapshot of the health of the ANS but only at the 24-hour level where it reflects circadian rhythm and has a lot more to do with sleep quality and the ability of the system to relax during sleep. 5-min or 15-min SDNN has nothing to do with this! More than that, the use of the LF/HF ratio and the concept of sympathovagal balance has been discredited. It is useful only to reflect sudden arousals. LF and HFnu have more validity. Also the report of no heart rate differences between groups makes no sense at all when comparing short term and 24-hour-derived values.

HRV is based on NN intervals and I question the assumption that RR means the same thing.

The authors do not have a deep understanding of HRV and the references seem to be selected without making sure they are current and meaningful. Here is an example:

"High HRV suggests dominant parasympathetic activity [12] with a good ability to adapt and respond to internal and external stimuli [3, 13]."

Ref 12 is from 1993, ref 3 is from 1996 and ref 13 is about autism. Actually high HRV from any recording but especially 24 hrs generally needs to be examined closely. Patients with atrial fibrillation have very high HRV and it is not a good sign. Patients can persent with an erratic sinus rhythm, where the ECG looks fine, and the irregularity associated with the rhythm can appear to be good vagal control of heart rate, but it is not.

Another example of references that are old or appear to have been thrown in without much thought:

Low HRV is an independent predictor of cardiac morbidity [11], due to dominant sympathetic activity [12].

Ref 11 is a comparison of hyperthyroid and euthyroid patients and unlikely to be a primary source for that statement and ref 12, again is from 1993 and a lot has changed. Again no distinction between 24-hr and very short term HRV.

Another quote:

Like a prism refracting light in its different wavelength components, the time domain can be separated in three components according to its frequency ranges [3]: low frequency (LF, 0.04 ± 0.15 Hz), high frequency (HF, 0.15 ± 0.4 Hz), and very low frequency (VLF, 0.003 ± 0.04 Hz).

HRV in the time domain is a purely statistical calculation and the assertion that the time domain can be separated into components is simply wrong. Moreover, for the purposes of this paper, VLF for the short recordings, is not likely to mean anything.

It is truly unfortunate that there is little or no literature about HRV in the non-linear domain in this field. Non-linear HRV tells us about the structure of the HR time series and allows us to distinquish between HRV that is normally organized and HRV that has an excess of disorganized patterns. HRV might be the same in the time domain and without actually examining the plots of the HRV power spectrum, the components, which are simply the area under the curve of power vs. frequency divided into bands, it is impossible to know if HRV is normal.

The review lumps HRV from digitized ECGs and HRV from devices that measure it from a pulse wave based device. Since pulse-based HRV does not have a clear R-wave for beat detection, they are not equivalent. Also, ectopic beats in an ECG-based analysis should be interpolated between the surrounding normal sinus beats, not excluded as stated in the paper.

I could go on, but let me get to the results, which are a noble attempt to summarize the findings but otherwise become boring and unreadable after a short time. There has to be a better way, but when apples are being treated as equivalent to oranges, in terms of how HRV was measured, it gets even worse. The English is good, but saying something to the effect that all studies found something and then after that saying, except two, does not work. It would work better if the authors said all but two studies, because then the reader does not feel like they are being told something "all studies" and then it is taken back.

I have a sense that the authors are highly skilled in actually doing a meta analysis, which is impressive, but really knowing the field and knowing how to cite references in a meaningful way is equally important and that is not here.

6. PLOS authors have the option to publish the peer review history of their article (what does this mean?). If published, this will include your full peer review and any attached files.

Reviewer #1: No

Reviewer #2: **Yes: **Phyllis K Stein

---

## [Author Response · Author response to Decision Letter 0]

28 Apr 2022

Academic Editor

Thank you for submitting your manuscript to PLOS ONE. After careful consideration, we feel that it has merit but does not fully meet PLOS ONE’s publication criteria as it currently stands. Therefore, we invite you to submit a revised version of the manuscript that addresses the points raised during the review process.

[REPLY] Thank you for your comment. We have addressed the comments of the reviewers in a revised manuscript and enclose a point-by-point response.

Please pay attention to all the reviewer's comments, especially those regarding potential bias and references and the use of digitized ECGs and HRVs from devices measuring it from a pulse wave based device. Use of the LF/HF ratio and current discussions regarding the utility of this should also be addressed.

[REPLY] Thank you for your comment. We have tried to respond to the reviewers' comments as best we can. Some references have been changed to more recent ones. All included articles were based on ECG measurements (Holter-ECG or ECG). No articles measuring pulse-based HRV were found in hypothyroidism. The limitations section now reads: "No included studies evaluated pulse-based HRV, which appears to be less accurate than ECG-based HRV [90]." Regarding the use of the LF/HF ratio, we have qualified its use in the limitations session. The limitations section now reads as follows: "The interpretation of the LF/HF ratio is controversial in the literature, and may not accurately reflect sympatho-vagal balance [91,92]. Ideally, the sympatho-vagal system tends more toward a nonlinear relationship [93,94]. We did not calculate a meta-analysis on the nonlinear assessment of HRV because it has been little studied in hypothyroidism. Parasympathetic-sympathetic interactions are complex, nonlinear, and often nonreciprocal [21]. Thus, nonlinear measures of HRV allow quantification of the unpredictability of a time series [92], which results from the complexity of HRV regulatory mechanisms [95-97]."

Journal Requirements

https://journals.plos.org/plosone/s/file?id=wjVg/PLOSOne_formatting_sample_main_body.pdf and https://journals.plos.org/plosone/s/file?id=ba62/PLOSOne_formatting_sample_title_ authors_affiliations.pdf 

[REPLY] We will adhere to the style requirements according to the style templates sent.

(This work was supported by the National Natural Science Foundation of China (Grant No. 51475317) and the Shanxi Provincial Natural Science Foundation of China (Grant No.201901D111237).)

Please provide an amended statement that declares *all* the funding or sources of support (whether external or internal to your organization) received during this study, as detailed online in our guide for authors at http://journals.plos.org/plosone/s/submit-now.  Please also include the statement “There was no additional external funding received for this study.” in your updated Funding Statement. Please include your amended Funding Statement within your cover letter. We will change the online submission form on your behalf.

[REPLY] The realization of this article has not received any funding. Our article is a meta-analysis, made from a systematic review of the literature. There should be a misunderstanding, or a wrong copy paste when you cite “(This work was supported by the National Natural Science Foundation of China (Grant No. 51475317) and the Shanxi Provincial Natural Science Foundation of China (Grant No.201901D111237)”. Those fundings does not belong to our article. 

(This work was supported by the National Natural Science Foundation of China (Grant No. 51475317) and the Shanxi Provincial Natural Science Foundation of China (Grant No.201901D111237).)

(This work was supported by the National Natural Science Foundation of China (Grant No. 51475317) and the Shanxi Provincial Natural Science Foundation of China (Grant No.201901D111237).)

[REPLY] We added the sentence: "Funding Statement: No funding has to be declared for the realization of this article". Our article is a meta-analysis, made from a systematic review of the literature, and received no specific funding.

[REPLY] The minimal data set underlying the results are described in the manuscript (S4 Figure). We were unable to access the data for each patient, but only for each included study.

5. Please amend the manuscript submission data (via Edit Submission) to author Bingbing Peng, and Pengcheng Liu.

[REPLY] Sorry, we do not understand who are Bingbing Peng and Pengcheng Liu. I think you may have made a mismatch or a wrong copy paste with a previous article submitted to Plos One. They are not among the authors of our article, and we do not know them …

6. Please amend your authorship list in your manuscript file to author Bingbing Peng, and Pengcheng Liu.

[REPLY] Bingbing Peng and Pengcheng Liu are not among the authors of our article … We do not know them … There should be a misunderstanding from your side.

Reviewer 1

This is an interesting systematic review and meta-analysis heart rate variability (HRV) in hypothyroid patients.

[REPLY] Thank you for your comment.

The conclusions from your review that HRV increases with severity of hypothyroidism are not particularly surprising. Much of the data from patients in the studies to which you refer were identified by routine blood test and were subclinical, their ‘hypothyroid status’ the result of deviation from the ‘normal values’.

[REPLY] Thank you for your comment. Indeed, we did not show that “HRV increases with severity of hypothyroidism” but we did show that “the alteration of HRV increases with the severity of hypothyroidism”. We have shown that this alteration exists when TSH is below 10mIU/L, as well as in cases of subclinical hypothyroidism. Therefore, we concluded a continuum between TSH levels and HRV alterations. 

Peixoto de Miranda et al., (2018) showed that subclinical hyperthyroidism leads to lower heart rate variability. These authors also found that no significant difference subclinical hypothyroidism group when compared to the euthyroid group. Peixoto de Miranda et al., (2018) therefore concluded that the subclinical thyroid dysfunctions presented no relationship with HRV variables. The conclusions from your paper seem to contradict this. Peixoto de Miranda et al., also adjusted for sociodemographic and clinical characteristics in their patients, and this is not apparent from the other references you have used in your review. Perhaps the negative findings from the Peixoto de Miranda et al. study are because the authors had adjusted for sociodemographic and clinical characteristics in their patients. It is well known that thyroid hormones and stress are linked in acute conditions.

[REPLY] Thank you for your comment. Patients included in the study by Peixoto de Miranda et al, (2018) had a small increase in TSH (5.1mIU/L on average), which may explain the lack of significant difference. We added the following sentence within the discussion: “Despite most included articles did not show HRV alteration depending on levels of TSH, we showed significant dose response relationship. It may be explained by the fact that each included article only retrieved a small increase in TSH levels, which may explain the absence of significant relation, whereas the combination of all articles in our meta-analysis permitted to analyze a wide range of TSH levels and HRV values.” We did not perform additional meta-regressions other than age, sex, BMI, blood pressure and thyroid function due to lack of data. On socio-demographic characteristics, only male gender is associated with lower LF/HF, which may be explained by lower sympathetic activity and higher parasympathetic activity compared to females (reference 77).

In [line 384] you say that "Men were associated with lower LF/HF ratio. This may be explained by the fact that men have lower sympathetic activity and higher parasympathetic activity compared to women". My first question when reading this was the expectation that there would therefore be a sex difference in HRV and hypothyroidism. It is disappointing to see that there is insufficient data in your review to investigate this further.

[REPLY] Thank you for your comment. Unfortunately, the data from the included studies did not allow us to investigate this issue further.

You don't seem to query the standard interpretation of the LF/HF ratio as sympathetic/ parasympathetic. It is not unequivocally accepted by everyone. (Billman, GE., (2013) The LF/HF ratio does not accurately measure cardiac sympatho-vagal balance. Front. Physiol., 20 February 2013 | https://doi.org/10.3389/fphys.2013.00026).

[REPLY] Thank you for your comment. The Limitations section now reads: “The interpretation of the LF/HF ratio is controversial in the literature, and may not correspond exactly to the sympatho-vagal balance [91,92]. Ideally, the sympatho-vagal system tends more towards a non-linear relationship [93,94].”.

[Line 425] Your conclusion is seriously deficient. Readers of this journal will be none the wiser by your statement about ‘deleterious effect of TSH on HRV parameters’. This is meaningless. There are a huge number of conditions that change HRV parameters that are not deleterious. Deleterious implies harm.

[Line 426] The statement ‘benefits of HRV assessment in the evaluation and monitoring of the severity of hypothyroidism should be further investigated’ is questionable – to what purpose? HRV can be affected by age, sex, stress, sociodemographic and clinical characteristics, and you have not given a breakdown due to lack of data.

[REPLY] Thank you for your comment. We removed the word deleterious and also reworded the conclusion. The conclusion now reads: “HRV is markedly decreased in hypothyroid patients. Increased sympathetic and decreased parasympathetic activity may be explained by molecular mechanisms involving catecholamines and by the effect of TSH on HRV parameters. The increased sympathetic and decreased parasympathetic activity may have clinical implications.”

I am happy with most of your interpretation of data. It is a shame that the lack of data precludes an adequate conclusion. It would help the reader if you were to include a list of abbreviations used in the paper. The world of HRV is difficult to understand by the novice.

[REPLY] Thank you for your comment. We added a new table 1:

HRV parameters

Acronym (unit)

Full name

Signification

Time-domain

 RR (ms)

RR–intervals (or Normal to Normal intervals – NN) i.e. beat-by-beat variations of heart rate

Overall autonomic activity

 SDNN (ms)

Standard deviation of RR intervals

Correlated with LF power

 RMSSD (ms)

Root mean square of successive RR-intervals differences

Associated with HF power and hence parasympathetic activity

 pNN50 (%)

Percentage of adjacent NN intervals varying by more than 50 milliseconds

Associated with HF power and hence parasympathetic activity

Frequency-domain

 TP (ms2)

Total power i.e. power of all spectral bands

Overall autonomic activity

 VLF (ms2)

Power of the Very Low Frequency (0.003 to 0.04 Hz)

Thermoregulation, renin-angiotensin system

 LF (ms2)

Power of the Low-Frequency band (0.04 to 0.15 Hz)

Index of both sympathetic and parasympathetic activity, with a predominance of sympathetic

 HF (ms2)

Power of the High-frequency band (0.15 to 0.4 Hz)

Represents the most efferent vagal (parasympathetic) activity to the sinus node

 LF/HF

LF/HF ratio

Sympathovagal balance

Reviewer 2

The idea of doing a meta-analysis of HRV and hypothyroidism that includes both short-term resting HRV and 24-hour recordings cannot be justified. SDNN from a 24-hour recording and SDNN from a 5-min recording cannot be compared and some of the short recordings even included paced breathing. SDNN is a powerful snapshot of the health of the ANS but only at the 24-hour level where it reflects circadian rhythm and has a lot more to do with sleep quality and the ability of the system to relax during sleep. 5-min or 15-min SDNN has nothing to do with this! […] Also the report of no heart rate differences between groups makes no sense at all when comparing short term and 24-hour-derived values.

[REPLY] Thank you for your comment. The effect sizes for each included article were calculated by comparing patients with hypothyroidism and controls within the same article (please see the methods section: “we conducted random effects meta-analyses (DerSimonian and Laird approach) for each HRV parameter comparing patients with untreated hypothyroidism with healthy controls [35].”) Therefore, we did not compare 5 minutes measures to 24 hours measures. First, we calculated the effect sizes for each included article. For example, we compared 5 minutes measures in untreated hypothyroidism patients and 5 minutes measures in healthy controls, in order to retrieve the corresponding effect size. On another article, we compared 24 hours measures in untreated hypothyroidism patients and 24 hours measures in healthy controls, in order to retrieve again the corresponding effect size. And so on for each included article. Therefore, even if we agree that differences in measurement time is a limitation, our data are valid as we pooled the effect sizes of each individual article to produce on overall results i.e. answering the question whether HRV is altered in patients with untreated hypothyroidism compared with healthy controls (and differences in measurement time are comprised within each effect sizes). The limitations section reads: “Studies also differed in measurement conditions, such as in duration of recording of HRV parameters [38, 46].”

More than that, the use of the LF/HF ratio and the concept of sympathovagal balance has been discredited. It is useful only to reflect sudden arousals. LF and HFnu have more validity.

[REPLY] Thank you for your comment. We totally agree with the discredit towards LF/HF ratio and the concept of sympathovagal balance. We have chosen to let the LF/HF ratio as it seems important to some readers. We added some references about the ongoing debate on the LF/HF ratio in the Limitations. The Limitations section now reads: “The interpretation of the LF/HF ratio is controversial in the literature, and may not correspond exactly to the sympatho-vagal balance [91,92]. Ideally, the sympatho-vagal system tends more towards a non-linear relationship [93,94].”.

HRV is based on NN intervals and I question the assumption that RR means the same thing.

[REPLY] Thank you for your comment. We added a new table 1 to explain the meaning of each HRV parameters. Yes, RR–intervals are also called Normal to Normal intervals – NN, i.e. beat-by-beat variations of heart rate.

The authors do not have a deep understanding of HRV and the references seem to be selected without making sure they are current and meaningful. Here is an example:

"High HRV suggests dominant parasympathetic activity [12] with a good ability to adapt and respond to internal and external stimuli [3, 13]." Ref 12 is from 1993, ref 3 is from 1996 and ref 13 is about autism.

[REPLY] Thank you for your comment. References were selected based on their quality, the reference from 1996 is the task force, still valid to date, and cited in all articles examining HRV. The reference on “Autism” from 2017 has in fact no link with autism, it is a vulgarization of the explanation of HRV. This article of 2017 was also judged sufficiently good to be posted in the Huffington post, for a mass communication. So the references were carefully chosen for readers to allow them both to have a deep understanding of HRV if they want to go in details, or to have a vulgarization for a quick understanding. Reference from 1993 is the first paper to give guidance on the interpretation of low or high HRV. Indeed, this reference is outdated, so we have removed it. For reference 11 of Ramanathan, this is indeed an error on our part. This reference has been replaced. The sentence now reads as follows: “Low HRV is an independent predictor of cardiac morbidity [11], while high HRV suggests good ability to adapt and respond to internal and external stimuli [3,12].”

Actually high HRV from any recording but especially 24 hours generally needs to be examined closely. Patients with atrial fibrillation have very high HRV and it is not a good sign. Patients can persent with an erratic sinus rhythm, where the ECG looks fine, and the irregularity associated with the rhythm can appear to be good vagal control of heart rate, but it is not.

[REPLY] Thank you for your comment. We totally agree with you. As it is a meta-analysis, we followed inclusion of included articles. Hyperthyroidism can indeed give erratic sinus rhythm and atrial fibrillation, but not in hypothyroidism.

Another example of references that are old or appear to have been thrown in without much thought:

Low HRV is an independent predictor of cardiac morbidity [11], due to dominant sympathetic activity [12]. 

Ref 11 is a comparison of hyperthyroid and euthyroid patients and unlikely to be a primary source for that statement and ref 12, again is from 1993 and a lot has changed. Again no distinction between 24-hr and very short term HRV.

[REPLY] Thank you for your comment. Reference from 1993 is the first paper to give guidance on the interpretation of low or high HRV. Indeed, this reference is outdated, so we have removed it. For reference 11 of Ramanathan, this is indeed an error on our part. This reference has been replaced. The sentence now reads as follows: “Low HRV is an independent predictor of cardiac morbidity [11], while high HRV suggests good ability to adapt and respond to internal and external stimuli [3,12].”

Another quote: Like a prism refracting light in its different wavelength components, the time domain can be separated in three components according to its frequency ranges [3]: low frequency (LF, 0.04 ± 0.15 Hz), high frequency (HF, 0.15 ± 0.4 Hz), and very low frequency (VLF, 0.003 ± 0.04Hz). HRV in the time domain is a purely statistical calculation and the assertion that the time domain can be separated into components is simply wrong. 

[REPLY] Thank you for your comment. Indeed, the time domain is a purely statistical calculation and is composed of RMSSD, SDNN, pNN50... We have also indicated this in the sentence: “In the time domain, the RR intervals (or normal-to-normal intervals-NN), the standard deviation of RR intervals (SDNN), the root mean square of successive RR-intervals differences (RMSSD) and the percentage of adjacent NN intervals varying by more than 50 milliseconds (pNN50) were analysed.”. As for the frequency domain, this is a picture to better imagine what the frequency domain is for HRV novices. But indeed, you are right, this sentence is confusing. We have changed the sentence to read as follows: “The frequency domain can be separated in three components according to its frequency ranges [3]: low frequency (LF, 0.04 ± 0.15 Hz), high frequency (HF, 0.15 ± 0.4 Hz), and very low frequency (VLF, 0.003 ± 0.04Hz).”

Moreover, for the purposes of this paper, VLF for the short recordings, is not likely to mean anything.

[REPLY] Thank you for your comment. Indeed, VLF is not measured in short recordings, and we have not performed a meta-analysis on VLF due to lack of data on 24-hour recordings.

It is truly unfortunate that there is little or no literature about HRV in the non-linear domain in this field. Non-linear HRV tells us about the structure of the HR time series and allows us to distinquish between HRV that is normally organized and HRV that has an excess of disorganized patterns. HRV might be the same in the time domain and without actually examining the plots of the HRV power spectrum, the components, which are simply the area under the curve of power vs. frequency divided into bands, it is impossible to know if HRV is normal.

[REPLY] Thank you for your comment. We totally agree. We added some references in favor of non-linear domain. The limitation section now reads: “We did not compute meta-analysis on non-linear assessment of HRV as it has been poorly studied in hypothyroidism. Parasympathetic-sympathetic interactions are complex, non-linear and often non-reciprocal [21]. Thus, non-linear measurements of HRV allow the unpredictability of a time series to be quantified [92], which results from the complexity of the HRV regulatory mechanisms [95–97].”

The review lumps HRV from digitized ECGs and HRV from devices that measure it from a pulse wave based device. Since pulse-based HRV does not have a clear R-wave for beat detection, they are not equivalent. Also, ectopic beats in an ECG-based analysis should be interpolated between the surrounding normal sinus beats, not excluded as stated in the paper.

[REPLY] Thank you for your comment. We totally agree that pulse-based HRV is less reliable than ECG measures. All articles included were based on ECG-measurements (Holter-ECG or ECG). No articles measuring pulse-based HRV was retrieved in hypothyroidism. The limitation section now reads: “No included studies assessed pulse-based HRV that seems to be less accurate than ECG-based HRV [90].”

I could go on, but let me get to the results, which are a noble attempt to summarize the findings but otherwise become boring and unreadable after a short time. There has to be a better way, but when apples are being treated as equivalent to oranges, in terms of how HRV was measured, it gets even worse. The English is good, but saying something to the effect that all studies found something and then after that saying, except two, does not work. It would work better if the authors said all but two studies, because then the reader does not feel like they are being told something "all studies" and then it is taken back.

[REPLY] Thank you for your comment. First we reworded some sentences, and second we take the opportunity to move the descriptive characteristics of included articles in supplementary materials. We now present within the main manuscript a summary of descriptive characteristics. Readers that still want further details can have access to them in supplementary materials. (S3 Figure) 

I have a sense that the authors are highly skilled in actually doing a meta analysis, which is impressive, but really knowing the field and knowing how to cite references in a meaningful way is equally important and that is not here.

[REPLY] Thank you for your comment. We removed some old historical references and we now cite more recent reviews or articles from high impact factor journals. We believe our article was strongly improved following your recommendations – particularly your suggestion to summarize the descriptive characteristics of included articles.

---

## [Decision Letter · Decision Letter 1]

18 May 2022

Heart rate variability in hypothyroid patients: A systematic review and meta-analysis

PONE-D-22-02213R1

Dear Dr. Brusseau,

We’re pleased to inform you that your manuscript has been judged scientifically suitable for publication and will be formally accepted for publication once it meets all outstanding technical requirements.

Kind regards,

Daniel M. Johnson, PhD

Academic Editor

PLOS ONE

Additional Editor Comments (optional):

Reviewers' comments:

Reviewer's Responses to Questions

**Comments to the Author**

1. If the authors have adequately addressed your comments raised in a previous round of review and you feel that this manuscript is now acceptable for publication, you may indicate that here to bypass the “Comments to the Author” section, enter your conflict of interest statement in the “Confidential to Editor” section, and submit your "Accept" recommendation.

Reviewer #1: All comments have been addressed

Reviewer #2: All comments have been addressed

2. Is the manuscript technically sound, and do the data support the conclusions?

Reviewer #1: Yes

Reviewer #2: Yes

3. Has the statistical analysis been performed appropriately and rigorously? 

Reviewer #1: Yes

Reviewer #2: Yes

4. Have the authors made all data underlying the findings in their manuscript fully available?

Reviewer #1: Yes

Reviewer #2: Yes

5. Is the manuscript presented in an intelligible fashion and written in standard English?

Reviewer #1: Yes

Reviewer #2: Yes

6. Review Comments to the Author

Reviewer #1: (No Response)

Reviewer #2: I belive this it now much improved, acceptable and worth publishing. However, mea culpa, I did not have the energy to again go thru it line by line to see if there was anything else :-( However, my comments were politely and responsively addressed.

7. PLOS authors have the option to publish the peer review history of their article (what does this mean?). If published, this will include your full peer review and any attached files.

Reviewer #1: **Yes: **Dr Duncan Banks

Reviewer #2: **Yes: **Phyllis K. Stein

---

## [Editor Report · Acceptance letter]

23 May 2022

PONE-D-22-02213R1 

Heart rate variability in hypothyroid patients: A systematic review and meta-analysis 

Dear Dr. Brusseau:

I'm pleased to inform you that your manuscript has been deemed suitable for publication in PLOS ONE. Congratulations! Your manuscript is now with our production department. 

Kind regards, 

on behalf of

Dr. Daniel M. Johnson 

Academic Editor

PLOS ONE